# Mesenchymal Stromal Cell Therapy in Lung Transplantation

**DOI:** 10.3390/bioengineering10060728

**Published:** 2023-06-17

**Authors:** Antti I. Nykänen, Mingyao Liu, Shaf Keshavjee

**Affiliations:** 1Latner Thoracic Surgery Research Laboratories, Toronto General Research Hospital Institute, University Health Network, Toronto, ON M5G 1L7, Canada; antti.nykanen@helsinki.fi (A.I.N.); mingyao.liu@utoronto.ca (M.L.); 2Temerty Faculty of Medicine, University of Toronto, Toronto, ON M5S 1A8, Canada; 3Department of Cardiothoracic Surgery, Helsinki University Hospital and University of Helsinki, FI-00029 Helsinki, Finland

**Keywords:** mesenchymal stromal cell, lung transplantation, cell therapy, gene therapy, ex vivo lung perfusion

## Abstract

Lung transplantation is often the only viable treatment option for a patient with end-stage lung disease. Lung transplant results have improved substantially over time, but ischemia-reperfusion injury, primary graft dysfunction, acute rejection, and chronic lung allograft dysfunction (CLAD) continue to be significant problems. Mesenchymal stromal cells (MSC) are pluripotent cells that have anti-inflammatory and protective paracrine effects and may be beneficial in solid organ transplantation. Here, we review the experimental studies where MSCs have been used to protect the donor lung against ischemia-reperfusion injury and alloimmune responses, as well as the experimental and clinical studies using MSCs to prevent or treat CLAD. In addition, we outline ex vivo lung perfusion (EVLP) as an optimal platform for donor lung MSC delivery, as well as how the therapeutic potential of MSCs could be further leveraged with genetic engineering.

## 1. Introduction

Lung transplantation is standard therapy for various end-stage lung diseases such as interstitial lung disease, obstructive lung disease, pulmonary hypertension, and cystic fibrosis. Almost 5000 lung transplants are reported annually in the International Society of Heart and Lung Transplantation Registry, and the results have improved over time [1]. However, the median survival of lung transplant recipients is limited to less than 7 years, and the main obstacles are primary graft dysfunction (PGD), acute rejection, chronic lung allograft dysfunction (CLAD), and side-effects of immunosuppressive drugs [2,3,4,5].

Mesenchymal stromal cells (MSC) are pluripotent cells present in most adult tissues [6,7]. MSC therapy in solid organ transplantation has gained attraction as MSCs possess anti-inflammatory, immunomodulatory, and protective features that could theoretically be utilized to counteract non-immunological and immunological injuries related to organ transplantation [8,9]. In lung transplantation, MSCs have been delivered in experimental studies to the donor lung to protect the transplant against ischemia-reperfusion injury (IRI) and PGD and to induce immunomodulatory effects [10]. In addition, small Phase 1 clinical trials have been performed in lung transplant patients with established CLAD [11,12,13], and additional clinical PGD and CLAD trials are ongoing [14].

Here, we review MSC biology, the most important pathological mechanisms involved in lung transplantation, and how these pathways could be targeted with MSC therapy. In addition, we describe how ex vivo lung perfusion (EVLP) could be used to administer MCSs into the donor lung ex vivo before transplantation [10,15], and highlight the potential of MSC genetic engineering to obtain enhanced therapeutic properties [16,17].

## 2. Mesenchymal Stromal Cells

Mesenchymal stromal cells (MSC) and multipotent adult progenitor cells (MAPC) are mesenchymal, non-hematopoietic progenitor cells that are found in most tissues and are capable of self-renewal and differentiation into various types of stromal cells (Figure 1). Resident MSCs are also present in the lung, localized in perivascular spaces, where they act as important regulators of pulmonary homeostasis and the balance between lung injury and repair [18]. For therapeutic purposes, MSCs are commonly isolated from bone marrow, adipose tissue, the umbilical cord, and the placenta [6,7]. As considerable heterogeneity in cell sources, and isolation and expansion methods exist, the International Society for Cell Therapy has recommended minimal standard criteria to identify and characterize MSCs. Accordingly, MSCs should adhere to plastic in standard culture conditions, express typical stromal cell surface markers, lack expression of hematopoietic markers, and differentiate into osteoblasts, adipocytes, and chondroblasts in vitro under appropriate conditions [19]. While MAPCs resemble MCSs, they are biologically more primitive and have greater differentiation potential than classical MSCs [20].

Initially, MSCs and MAPCs were considered to result in cell replacement and renewal of damaged tissues; however, the current view is that the therapeutic effects of these cells are largely mediated by paracrine signaling, through the secretion of numerous soluble mediators, including growth factors, cytokines, and microRNA. This MSC secretome is considered to modulate inflammation and immune responses, and to stimulate innate tissue repair [6,21]. MSC therapies have been investigated in numerous clinical trials of cardiovascular disease, graft-versus host disease, and neurological disease, but the success has so far been variable [7]. Several studies have also examined the use of MSC in lung disease, especially in acute respiratory distress syndrome (ARDS) [22,23]. Here we will concentrate on the use of MSCs in lung transplantation.

### 2.1. MSC Surface Receptors and Immunomodulatory Effects

According to the minimal standard criteria, MSCs express stromal cell markers CD105, CD73, and CD90 and lack the expression of hematopoietic surface molecules CD45, CD34, CD14 or CD11b, CD79alpha or CD19, and HLA-DR in vitro [19]. MSCs are considered immunoprivileged as they have HLA-I antigens but lack the expression of HLA-II and the costimulatory molecules/receptors CD80 and CD86 that are needed for T cell activation. This has important therapeutic implications for MSCs, as allogenic MSC may be used without eliciting a robust immunoreaction or cytolysis [6,17].

Several studies indicate that MSCs have immunomodulatory effects. They suppress mixed lymphocyte reactions in vitro and alloimmune reactions in vivo through various mechanisms [17]. Similarly, resident MSCs isolated from human lung transplants inhibit T cell activation in vitro, indicating that MSCs may mediate immunological responses in an allogeneic lung environment [24,25]. MSCs interact with numerous innate and adaptive immune cells, both through direct contact with the target cells and also via various paracrine factors [17]. Important for solid organ transplantation, MSCs polarize naïve and memory CD4^+^ T cells towards regulatory T cells (Treg) and promote tolerance and long-term allograft acceptance. Additionally, MSCs induce the formation of regulatory CD8^+^ T cells and regulatory B cells. MSCs also have suppressive effects on antigen presenting cells such as dendritic cells and induce polarization of macrophages from the pro-inflammatory M1 to the anti-inflammatory M2 subtype [6,17,26].

### 2.2. MSC Secretome and Paracrine Effects

Several cytokines and pathways are considered important for the anti-inflammatory and immunomodulatory effects of MSC [21]. One of the most important factors is indoleamine 2,3-dioxygenase (IDO), an enzyme catalyzing the breakdown of tryptophan that induces immunosuppression through the generation of Tregs and tolerogenic dendritic cells (DC) [17,27]. Another key immunomodulatory molecule is HLA-G. MSCs express both the membrane-bound HLA-G1 and soluble HLA-G5 forms, which are considered to polarize T helper cells into Th2 cells [17]. An additional important molecule is leukemia inhibitory factor (LIF), which is expressed at high levels by MSCs and induces direct inhibition of effector T cells and promotes the generation of Tregs [17]. Several other secreted factors also play a role in the immunomodulatory effects of MSCs (Figure 1) [6,21]. It has been suggested that two different immunoregulatory networks may be essential, with the induction of tolerogenic genes IDO, HLA-G, and LIF functioning in a contact-independent manner and direct links with MSCs and target cells leading to modulation of IL-10 and TGF-β expression by the T cells [28].

Although MSCs from different sources are similar, their protein, transcriptomic, and secretory profiles vary to some extent. These differences likely reflect distinctive biological properties and indicate that it is important to consider the MSC source when planning, predicting, or interpreting possible therapeutic effects [17,21].

### 2.3. MSC-Derived Extracellular Vesicles

Extracellular vesicles (EVs) are lipid bilayer particles released from various cells, such as MSCs that are considered important for cell-to-cell interactions. EVs contain lipids, nucleic acids, and proteins and are categorized as exosomes, microparticles, and apoptotic bodies based on the EV size. Interestingly, several studies have shown that administration of EVs derived from MSCs, rather than the MSCs themselves, protects against lung injury [23,29,30,31]. These results indicate that EVs contain bioactive components that are important for intercellular communication and may in fact mediate many of the beneficial effects of MSCs [23,31]. Consequently, there has been considerable interest in developing cell-free EV-based therapies that could elicit therapeutic effects without some of the potential concerns of MSCs such as oncogenesis or intravascular clustering and embolization [23].

### 2.4. MSCs in Lung Injury

MSCs protect lung epithelial cells in vitro through paracrine mechanisms by downregulating inflammatory and upregulating anti-apoptotic factors [32,33] and promoting epithelial repair [34]. MSCs also have protective effects on pulmonary endothelial cells by enhancing endothelial barrier function and inhibiting endothelial-to-mesenchymal transition [35,36,37,38]. Additionally, effects on alveolar macrophages, which constitute the first-line defense for the lung, are considered important [26]. When given systemically, MSCs home to the site of damage: this involves an interplay of chemokine signals and adhesion molecules and is considered one of the advantages of using MSCs in lung injury [22]. Interestingly, the lung microenvironment may also affect MSC properties, as the MSC secretome can change, for example, due to inflammation or acidic conditions present in severe lung injury [35,39,40].

MSCs have been shown to be beneficial in various experimental models of lung injury and disease [10,23,31], and these effects likely involve antimicrobial, anti-inflammatory, regenerative, anti-oxidative stress, angiogenic, and antifibrotic signals [41]. Several clinical studies have investigated the safety and efficacy of MSC therapy in ARDS [41]. A recent meta-analysis of 13 clinical randomized controlled trials with a total of 655 patients using intravenous administration of MSCs in ARDS indicated that MSC therapy is safe and may decrease mortality [22]. However, the majority of these studies were performed in patients with COVID-19, and only 2 studies were on non-COVID-19 ARDS. Therefore, more large-scale clinical trials are needed to confirm the efficacy of MSC therapy in ARDS [22].

## 3. Lung Transplant MSC Therapy

The protective anti-inflammatory and immunomodulatory properties make MSCs attractive therapeutic candidates for lung transplantation. MSCs have been delivered to the lungs through the airways or intravascularly and can be administered into the donor lung either before transplantation or to the recipient before or after transplantation (Figure 2). Most of the preclinical studies have used MSCs to inhibit IRI, while some experiments have targeted acute rejection, and some clinical trials have explored MSC therapy in patients with CLAD (Table 1). Importantly, there is significant interaction between IRI, acute rejection, and CLAD (Figure 3), as lung injury may trigger alloimmune responses, and various non-immunological and immunological factors participate in the development of CLAD [5,42].

### 3.1. MSC Therapy in Lung Transplant IRI and PGD

PGD occurs in almost one-third of lung transplant recipients [65]. Severe PGD is especially problematic as mortality is high and treatment options are mainly supportive [66,67,68]. The clinical importance of PGD is further emphasized by its negative long-term effects [69,70]. Overproduction of reactive oxygen species after ischemia-reoxygenation is considered the main trigger in IRI and PGD [71,72,73,74,75,76]. This results in widespread molecular and cellular dysfunction that involves cytokines and damage-associated molecular patterns (DAMP), innate and adaptive immunity activation, mitochondrial dysfunction, ion imbalance, apoptosis, and the activation of complement, coagulation, and platelet-aggregation cascades. The ensuing microvascular dysfunction and breakdown of the alveolar-capillary barrier result in pulmonary edema and hypoxia—the clinical manifestations of PGD [71,72]. The main cells involved in IRI consist of both parenchymal and inflammatory cells. Endothelial and epithelial cells, and donor alveolar macrophages, are considered essential contributors to the initial IRI [67], and danger signals from these cells augment the infiltration of recipient monocytes, macrophages, neutrophils, natural killer cells [76,77], and lymphocytes.

Several experimental studies have shown that MSC therapy is beneficial for lung IRI (Table 1). The majority of these studies have used bone marrow-derived cells, but also MSCs from the umbilical cord or adipose tissue have also been effective. Airway or intravascular delivery results in alveolar or blood vessel MSC localization, respectively [53], and intravascular delivery has been shown to yield better MSC retention in the lung parenchyma in a pig EVLP model [48]. Multiple studies indicate that MSC therapy protects the alveolar-epithelial barrier during lung injury, as improved permeability, alveolar fluid clearance, and oxygenation have been reported (Table 1). Anti-inflammatory effects are likely also important, as MSC therapy decreases proinflammatory cytokine levels and neutrophilia [10,15,45,46,48,52,55,58,59,78]. Several studies have also reported that administration of extracellular vesicles (EV) derived from MSCs instead of MSCs during EVLP protects against lung injury [29,30]. No clinical lung transplant IRI studies have been reported yet, but one Phase 1–2 trial is investigating the effect of adipose tissue-derived MSCs on PGD (ClinicalTrials.gov identifier NCT04714801).

### 3.2. MSC Therapy in Acute Rejection

Acute rejection remains a significant problem after lung transplantation, and about 27% of the recipients experience at least one acute rejection episode during the first year [79]. Acute rejection most often responds to anti-rejection treatments, but recurrent rejection episodes are problematic and may increase the risk of the later development of chronic rejection [3,80]. In addition, the augmented immunosuppression predisposes the patients to infections, malignancies, and metabolic problems [81]. Allorecognition after solid organ transplantation occurs in lymph nodes or the spleen, but in the case of the lung, it can also take place locally in the transplanted graft itself [82,83]. Co-stimulatory signals and cytokine-mediated lymphocyte differentiation and expansion are also needed to mount an efficient alloimmune response, and the effector mechanisms involve CD4^+^ helper T cells, CD8^+^ cytotoxic T cells, B cells, antibody production, and complement-mediated damage. Importantly, Tregs function to limit the alloimmune response [84].

MSC delivery into the lung graft to prevent or treat acute rejection would be attractive based on the immunomodulatory properties of MSCs. However, although MSC therapy has early protective effects in lung transplant IRI (Table 1), it remains largely unknown whether this reflects decreased alloimmunity later after transplantation. In a rat major histocompatibility complex (MHC)-mismatched single lung transplant model, MSC administration on days 0 and 3 decreased histological signs of rejection and resulted in a beneficial change in PD-L1 and IL-17A gene expression at 6 days [60]. Similarly, in a mouse MHC-mismatched orthotopic tracheal transplant model with follow-up until 90 days, human induced pluripotent stem cell (iPSC)-derived MSCs resulted in immunomodulation and an increase of Tregs [61]. These experimental studies indicate that MSCs can inhibit the development of alloimmune responses, but more translational experiments and clinical trials are needed. In addition, it remains to be determined whether MSCs could also be used, for example, as salvage therapy to treat established or recurrent acute rejection episodes.

### 3.3. MSC Therapy in CLAD

CLAD results in progressive and irreversible deterioration of graft function and remains the leading cause of late deaths after lung transplantation [85,86]. CLAD currently consists of different subtypes. Bronchiolitis obliterans syndrome (BOS) is characterized by obstruction and small airway obliteration [85], whereas restrictive allograft syndrome (RAS) results in restrictive lung function and fibrosis [87]. Additionally, mixed and undefined phenotypes are recognized [88,89]. Therapeutic options for CLAD are limited, and it is known that the prognosis of RAS is poor [85,90]. These factors highlight the need for novel therapeutic options to prevent CLAD development or treat established diseases. It is believed that chronic non-immunological and immunological injury to the lung is central to causing pathological distal airway fibroproliferation, leading to the irreversible lung damage related to CLAD [42,86,91,92]. Encouragingly, in mouse tracheal transplantation models, MSC therapy has successfully prevented airway obliteration, possibly by modulating immune responses [61,62,63,64].

MSCs have also been used in three important Phase 1 clinical trials that have established the safety and feasibility of MSC therapy in lung transplant patients [11,12,13]. All of these studies have used cryopreserved 3rd party allogeneic bone marrow-derived MSCs with intravenous administration to patients with obstructive CLAD, i.e., BOS. Chambers et al. delivered 2 × 10^6^ MSCs/kg to 10 lung transplant recipients with BOS grade ≥ 1, or to patients with grade 1 with risk factors for CLAD progression. The treatment was feasible and safe; no procedure-related serious adverse effects were detected, and there was a trend in the lung function decline measured by forced expiratory flow in 1 s (FEV_1_) was smaller after MSC infusion compared with the pre-treatment situation [13]. Keller et al. used a single intravenous infusion in nine patients, comparing three doses of 1, 2, or 4 × 10^6^ MSCs/kg. MSC therapy was feasible, safe, and well tolerated and did not change functional or laboratory parameters within 30 days after infusion [12]. Erasmus et al. treated 13 patients with moderate-to-severe BOS using 0.5 or 1 × 10^6^ MSCs/kg. Lung function decline is measured by FEV_1_ was significant before treatment but not after treatment, indicating possible CLAD stabilization [11]. An ongoing Phase 2 trial investigates whether 4 intravenous doses of 2 × 10^6^ MSCs/kg given over 2 weeks improve CLAD-progression-free survival (ClinicalTrials.gov identifier NCT02709343). The results of this study will hopefully help clarify the therapeutic potential of MSCs in lung transplantation. In addition, more studies are needed to determine the optimal MSC source and dose, CLAD severity, and possible effects in other CLAD subtypes than BOS.

## 4. Strategies to Improve MSC Delivery and Therapeutic Potential

Although the lung is well accessible for MSC therapy with either airway or intravenous in vivo administration, cell delivery into an isolated donor lung during ex vivo lung perfusion (EVLP) would have several advantages. We review the concept of EVLP and how it could be used as a unique therapeutic opportunity and platform for MSC therapy. In addition, we discuss how MSC genetic engineering could be used to improve cell properties and achieve improved therapeutic efficacy.

### 4.1. MSC Delivery into the Donor Lung during EVLP

In EVLP, the isolated donor lung is perfused and ventilated in normothermia ex vivo (outside the body), enabling confirmation of donor lung quality before transplantation. This has revolutionized lung transplantation, as marginal donor lungs that would otherwise have been rejected from transplantation can be tested and transplanted if graft function is adequate [93]. A landmark clinical study evaluated high-risk donor lungs during EVLP and established that transplantation of lungs that were physiologically stable during 4 h of EVLP leads to post-transplant results similar to those obtained with conventionally selected lungs [94]. EVLP has safely and significantly increased the available pool of donor organs for transplantation, and later studies have shown that medium- and long-term lung transplant results remain excellent after EVLP evaluation [95,96].

EVLP has been used as a platform for cell therapy in several experimental studies using airway or intravascular delivery (Table 1). Mordant et al. compared different MSC doses and administration routes during pig lung EVLP. Intravascular administration resulted in better MSC retention in the lung, and most of the MSCs engraftment occurred within the first minutes after pulmonary artery injection [48]. Moreover, comparison of three different doses revealed that intravascular administration of 150 × 10^6^ MSCs, the middle dose of the experiment, improved lung oxygenation and compliance. In contrast, a higher 300 × 10^6^ MSCs dose temporarily increased pulmonary vascular resistance [48], possibly related to clustering of the cells into the lung microvasculature. These results highlight the efficacy of intravascular MSC delivery but also indicate that high cell doses may result in adverse embolic effects [48]. Interestingly, the optimal cell dose (150 × 10^6^ MSCs; 5 × 10^6^ per kg of animal weight) is close to the doses used in the clinical MSC CLAD studies [11,13,70]. In the studies administering MSCs during EVLP, decreased proinflammatory cytokine levels and protection of the alveolar-epithelial barrier function during ischemia- or endotoxin-induced lung injury have been observed [45,46,48,52,55,58,59]. Several studies have also reported that administration of MSC-derived EVs protects against lung injury [29,30].

### 4.2. Genetically Engineered MSCs

Genetic engineering of cell-based therapeutics with viral and non-viral vectors has been used to achieve modified cells with enhanced therapeutic properties. Genetic engineering modulates the MSC secretome and aims, for example, to enhance MSC migration to target sites, facilitate intrinsic repair properties, improve cell survival, or change MSCs into gene delivery vehicles [16].

Several different genetically modified MSCs have been studied in lung injury models, and for example, MSC engineered to produce anti-inflammatory IL-10 have been beneficial in lung IRI, endotoxemia, and ARDS [43,97,98]. In our recent study, we transduced human umbilical cord-derived MSC with adenoviral vectors encoding IL-10 and administered these pre-modified and cryopreserved MSC^IL−10^ cells during EVLP to human lungs rejected from clinical transplantation. This rapidly elevated EVLP perfusate IL-10 levels in minutes, MSC^IL−10^ cells were retained in the lung, and IL-10 was increased also in lung tissue and the airway compartment [40]. While possible therapeutic or immunomodulatory effects of MSC^IL−10^ therapy are not known yet, these results indicate that therapy with genetically modified cells during standard clinical EVLP is feasible.

## 5. Conclusions

MSCs have several anti-inflammatory, immunomodulatory, and repair capabilities that make them attractive therapeutic options in lung transplantation. Experimental studies show that perioperative MSC administration protects the donor lung against IRI, acute rejection, and the development of CLAD mainly through paracrine mechanisms. As these conditions remain major obstacles to the long-term survival of lung transplant patients, novel therapeutic options are indeed warranted. However, although small clinical trials indicate that MSC therapy is safe and feasible in patients with established CLAD, randomized studies are needed to confirm therapeutic efficacy. In addition, the optimal MSC source, dose, and preparation need to be determined. It will also be important to define the potential role of cell-free MSC-derived EVs. In the future, it may be possible to use EVLP to deliver genetically modified MSCs with improved properties to achieve advanced lung transplant cellular engineering for enhanced protection and immunomodulation.

## Figures and Tables

**Figure 1 bioengineering-10-00728-f001:**
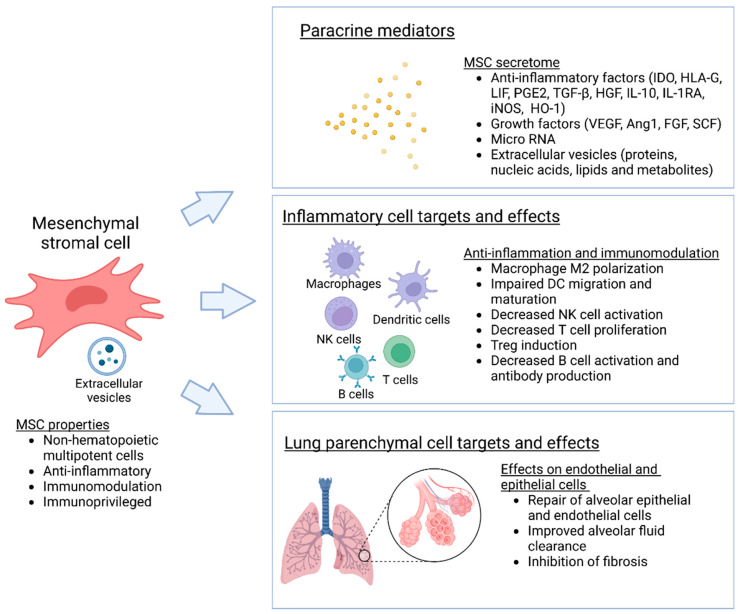
Biological properties and effects of mesenchymal stromal cells. MSC are non-hematopoietic multipotent cells that have anti-inflammatory, immunomodulatory and protective effects mainly mediated though paracrine mechanisms.

**Figure 2 bioengineering-10-00728-f002:**
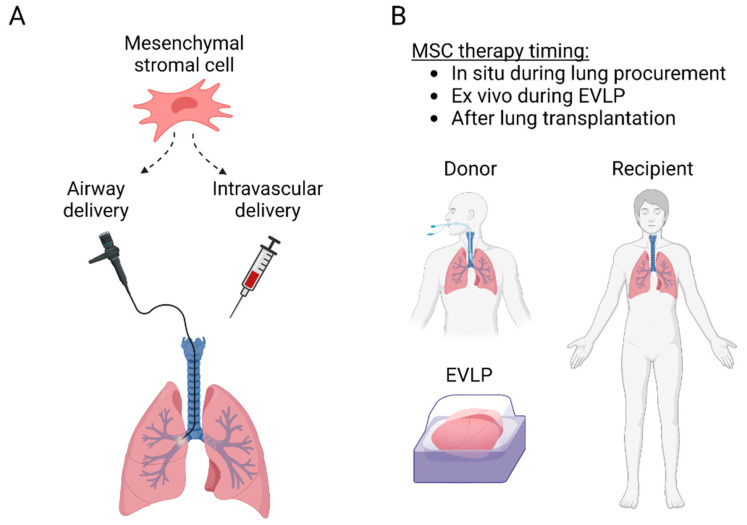
(**A**) MSC delivery routes to the lung transplant. MSC can be administered into the lungs through the airways with a flexible bronchoscope or intravascularly either intravenously, or directly through the pulmonary artery. (**B**) MSC administration can be performed before transplantation in situ in the donor or ex vivo during ex vivo lung perfusion, or after transplantation to the recipient.

**Figure 3 bioengineering-10-00728-f003:**
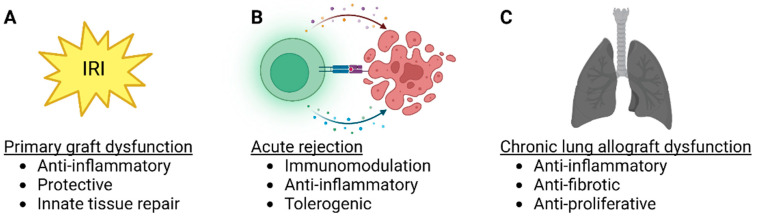
MSC therapy targets in lung transplantation. (**A**) Lung transplant ischemia-reperfusion injury and primary graft dysfunction, (**B**) acute rejection and (**C**) chronic lung allograft dysfunction (CLAD) are potential targets for MSC therapy. In each lung transplant pathology, possible beneficial effects of MSCs are listed.

**Table 1 bioengineering-10-00728-t001:** Lung transplant and EVLP studies with MSCs.

Target	Species	Model	MSC	Administration	Effect	Ref
IRI	Rat	Lung hilar clamping	Engineered BM-MSC (MSC-vIL-10)	Intravenous	Improved oxygenation, inflammation and permeability	[43]
IRI	Pig	SLTx	BM-MSC	Pulmonary artery vs. endobronchial	Endobronchial MSC delivery improved lung compliance	[44]
IRI	Human	EVLP	MAPC	Airways	Decreased edema and inflammation	[45]
IRI	Human	EVLP	BM-MSC	Intravascular	Restored alveolar fluid clearance	[46]
IRI	Mouse	SLTx	BM-MSC	Recipient intravenous	Decreased IRI, MSC homing preferentially into the lung transplant	[47]
IRI	Pig	EVLP	UC-MSC	Airway vs. intravascular, 3 different doses	Intravascular delivery improved MSC lung retention, optimal dose 150 × 10^6^ MSC decreased IL-8 and increased VEGF	[48]
IRI	Mouse	SLTx	BM-MSC	Ex vivo pulmonary artery	Decreased IRI	[49]
IRI	Pig	SLTx	BM-MSC	Pulmonary artery vs. endobronchial	No short-term differences detected	[50]
IRI	Mouse	Lung hilar clamping and EVLP	Human UC-MSC vs. MSC-EVs	Intravascular	MSCs and MSC-EVs attenuate IRI	[51]
IRI	Human	EVLP	MAPC	Airways	Decreased BAL neutrophilia, TNF-α, IL-1β and IFN-γ	[52]
IRI	Pig	SLTx	BM-MSC	Intravenous or intrabronchial	Heterogenous localization, in alveoli after endobronchial and in blood vessels after intravascular administration	[53]
IRI	Rat	SLTx	BM-MSC	Intravenous	Protection against IRI	[54]
IRI	Pig	EVLP and SLTx	UC-MSC	Intravascular	Decreased IRI during EVLP and after TX	[55]
IRI	Rat	EVLP	UC-MSC	Intravascular	Improved inflammation and IRI	[56]
IRI	Rat	EVLP	BM-MSC-EVs	Intravascular	Multiple influences on pulmonary energetics, tissue integrity and gene expression	[30]
IRI	Human	EVLP	Engineered UC-MSC (MSC^IL−10^)	Intravascular	Safe and feasible, results in rapid IL-10 elevation	[40]
IRI	Rat	SLTx	Donor vs. recipient adipose tissue MSC	Intravenous	MSCs, regardless of their origin, exert similar immunosuppressive effects	[57]
IRI/ ARDS	Human	EVLP/endotoxin	BM-MSC	Airways	Restored alveolar fluid clearance	[58]
IRI/ ARDS	Human	EVLP/e.coli pneumonia	BM-MSC	Airways	Restored alveolar fluid clearance, reduced inflammation and increased antimicrobial activity	[59]
Acute rejection	Rat	SLTx	BM-MSC	1 vs. 2 recipient intravenous doses	Protection from acute rejection, best result with 2 recipient doses	[60]
Acute rejection/ CLAD	Mouse	Ortotopic tracheal Tx	iPSC-MSC	Intravascular	Induces immune tolerance and supports long-term graft survival	[61]
CLAD	Mouse	Heterotopic tracheal Tx	MSC (various sources)	Intravenous	Prevents airway occlusion	[62]
CLAD	Mouse	Ortotopic tracheal Tx	BM-MSC	Intravenous	Prevents airway occlusion through macrophage cytokines	[63]
CLAD	Mouse	Heterotopic tracheal Tx	BM-MSC	Local vs. systemic vs. combination	Prevents airway occlusion through modulation of immune response, best effect with combination treatment	[64]
CLAD	Human	Clinical Tx	BM-MSC	Intravenous twice weekly for 2 weeks	Safe and feasible in patients with advanced CLAD	[13]
CLAD	Human	Clinical Tx	BM-MSC	Intravenous	Safe and feasible in patients with moderate CLAD	[12]
CLAD	Human	Clinical Tx	BM-MSC	Intravenous	Well tolerated in moderate-to-severe CLAD, low-dose may slow progression of CLAD in some patients	[11]

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
