# Peer review of "Mesenchymal Stromal Cell Therapy in Lung Transplantation"

_bioengineering, 2023, doi:10.3390/bioengineering10060728_

Round 1
Reviewer 1 Report
I very much enjoyed reading this review and have no suggestions that would improve its clarity of message. It is comprehensive and well organised.
Author Response
We would like to thank the Reviewer for taking the time to review our work and for the comments.
Reviewer 2 Report
1. The title reflects the content of the article.
2. In the "Abstract" section, the authors briefly presented the content of this review article. So, the problems of lung transplantation are indicated. The effect of mesenchymal stromal cells (MSCS) is characterized. The therapeutic possibilities of MSCS therapy in lung transplantation are presented.
3. All the "Keywords" presented in the article are necessary.
4. In the section "1. Introduction" the authors described lung transplantation, which is a standard therapy for various terminal stages of lung diseases, and complications during its use. The design of the article is presented: the authors examined the properties of MSCs useful for transplantation, and the pathological mechanisms involved in lung transplantation, as well as ways to change them using MSCS therapy.
The article is presented in such sections as "2. Mesenchymal stromal cells", "3. Lung transplant MSC therapy", "4. Strategies to improve MSC delivery and therapeutic potential", "5. Conclusions". The section "2. Mesenchymal stromal cells" describes the heterogeneity of cell sources, as well as methods of isolation and reproduction of MSCs, recommended minimum standard criteria for the identification and characterization of MSCs, therapeutic effects and mechanisms of action of MSCs. It is indicated that the methods of treating MSCs have been investigated in numerous clinical trials. In the section "3. Lung transplant MSC therapy" describes the methods of delivery of MSCs to the lungs, the duration of MSCs therapy, the therapeutic effects of MSCs, and complications of lung transplantation in various diseases. In the section "4. Strategies to improve MSC delivery and therapeutic potential", the authors described possible strategies for improving the delivery of MSC to the lungs and discussed MSC genetic engineering approaches to improve cell properties and achieve increased therapeutic efficacy.
5. In the "Conclusion" section, the authors summarized their analysis of the literature. The authors noted the most promising strategies for improving MSC-therapy. In particular, the authors pointed out that the perioperative administration of MSCs protects the donor lung from ischemia-reperfusion injury (IRI), acute rejection and the development of lung allograft dysfunction (CLAD) mainly due to paracrine mechanisms. It is necessary to solve such issues as determining the optimal source of MSCs, dose and preparation need to be determined, choosing optimal methods for genetic modification of MSCs in order to improve their properties, protection and immunomodulation. The presented conclusion fully corresponds to this review article.
6. The text of the article is written clearly, accompanied by figures and tables that are necessary.
7. The article is interesting, important and timely, does not cause any concerns. The manuscript did not cause any ethical problems. All references to publications presented by the authors in the article are necessary and correct, made in the right style. I have no concerns about the similarity of this article with other articles published by the same authors. All figures and tables are necessary, understandable.
8. Competing interests of authors do not create bias in the presentation of results and conclusions.
Author Response

(The authors gave the same response as above.)
